# Stress Response and Safe Driving Time of Bus Drivers in Hot Weather

**DOI:** 10.3390/ijerph19159662

**Published:** 2022-08-05

**Authors:** Xianglong Sun, Junman Dong

**Affiliations:** School of Civil Engineering, Northeast Forestry University, Harbin 150040, China

**Keywords:** bus driver, high temperatures, stress response, safe driving time, survival rate

## Abstract

Purpose: To evaluate the impact of high-temperature environments on bus drivers’ physiology and reaction times, and to provide a basis for driver occupational health management. Methods: The physiological and reaction indexes of 24 bus drivers under different temperatures were investigated. The statistical analysis method was used to analyze the changes in drivers’ physiological stress, the relationship between stress and response ability, and a safe driving time. The Kaplan–Meier survival function was used to analyze the survival rate of bus drivers under different temperatures and driving times. Results: The results showed that body temperature, heart rate, physiological strain index (*PSI*), and reaction ability were significantly different among different compartment temperatures. *PSI* was positively correlated with reaction ability. The safe driving time was 80 min, 73 min, and 53 min, respectively, at 32 °C, 36 °C, and 40 °C. The survival rate decreased to less than 60% at 36 °C when driving continuously for 73 min; it decreased to 20% at 40 °C when driving for 53 min, and it was 0 for 75 min. Conclusions: High-temperature environments lead to heat stress of bus drivers, physiological indexes have changed significantly, and behavioral ability is also decreased. The higher the temperature, the lower the survival rate. Improvement measures can be taken from the aspects of convection, conduction, and behavior to ensure the bus driver’s physiological health and driving safety under high temperatures and to improve the survival rate.

## 1. Introduction

With global warming, high-temperature weather appears more frequently. The impact of high temperatures on bus drivers and passengers has attracted increasing attention. Presently, the proportion of air-conditioned buses in China’s big cities is about 70%. However, there are still many non-air-conditioned buses in use, and the proportion is higher in some small and medium-sized cities. For non-air-conditioned buses, the compartment temperature is 5–10 °C higher than the air temperature in the summer. When the air temperature reaches 32 °C, the compartment temperature can be as high as 40 °C. It is called a high-temperature environment when the living environment exceeds 35 °C, or the work environment exceeds 32 °C. Bus drivers exposed to high temperatures for a long time will suffer heat stress. Heat stress refers to the total nonspecific physiological response of the body to the external thermal environment. The changes in body temperature, heart rate, blood pressure, and sweating rate caused by working at high temperatures are attributed to heat stress [1]. 

There are many studies on the effects of high temperature on physiology. The researchers measure the physiological parameters of soldiers [2], athletes [3,4,5], construction workers [6,7], and farmers [8,9], observe their change trends, and determine a series of evaluation indexes such as thermal stress and thermal limit. High temperature can also affect the driver’s behavioral response. There is a significant positive correlation between thermal comfort and the driving ability of bus drivers [10]. Heat stress hurts the driver’s alertness, distracting the driver’s attention and leading to overt driving errors [11,12,13].

Currently, the research on stress response to high temperatures mainly focuses on athletes, workers, farmers, soldiers, and car drivers. In contrast, as a group working in high temperatures, bus drivers receive little attention. Therefore, it is significant to determine the relationship between temperature, driving time, and physiological indicators for bus driver occupational health management.

This study will investigate the bus drivers’ physiological indexes and behavioral responses under different temperatures and analyze their changing rules. We calculate the safe driving time based on the physiological safety limit and analyze the driver’s survival risk using a risk analysis model. 

## 2. Evaluation of Physiological Stress and Behavioral Response

### 2.1. Physiological Stress

When people are in a state of stress, some physiological indexes are significantly affected. Moran et al. propose a physiological strain index (*PSI*) based on rectal temperature (*RT*) and heart rate (*HR*) [14]. *PSI* describes the effects of environment and labor on the human cardiovascular system and thermoregulation system. The weight of *RT* and *HR* is the same in the calculation, and the formula is as follows:$$PSI=5\frac{HR_{t}-HR_{0}}{180-HR_{0}}+5\frac{RT_{t}-RT_{0}}{39.5-RT_{0}}$$
where *HR_t_* (*RT_t_*) is the heart rate (rectal temperature) at time *t*; *HR*_0_ (*RT*_0_) is the heart rate (rectal temperature) at time 0, which is the initial value of the human body before high-temperature work. 180 indicates that the upper limit of the heart rate is 180 bpm; 39.5 indicates that the upper limit of rectal temperature is 39.5 °C.

The Bureau of International Labor Affairs proposes the safe values of heart rate and rectal temperature under different labor intensities, as shown in Table 1.

In order to grade *PSI*, we take 100 bpm as *HR*_0_ and 37.5 °C as *RT*_0_. The classification of *PSI* is shown in Table 2. 

During work, when *PSI* is less than 2.8, it means that the physiological index of the worker is normal; when *PSI* is between 2.8 and 5.6, it means that the physiological index exceeds the standard, and appropriate rest should be taken. When *PSI* is greater than 5.6, the worker should stop working immediately and have a rest.

### 2.2. Behavioral Response

The driver easily becomes agitated at high temperatures, leading to unreasonable driving choices. Two indexes of “Choice reaction time” and “the number of choice errors” are used to evaluate response ability [15]. The former is to check the stability of the driver’s handling ability in the face of more than two complicated situations. The latter is to check whether the driver accurately processes more than two kinds of complex traffic information.

## 3. Method

### 3.1. Measurement Method

(1)Heart rate

Heart rate is the number of heartbeats per minute (bpm). Heat stress and cardiovascular tension caused by changes in labor intensity and ambient temperature can be directly reflected in the increase in heart rate. Compared with other physiological indexes, HR is more sensitive to the environment and easy to measure. It is widely used in thermal stress evaluation. The normal range is 60–120 bpm, while in a high temperature and high humidity environment, the limit of the heart rate is 180 bpm. Usually, the measurement method of heart rate is relatively simple, and a heart rate sensing wristband can obtain the heart rate data. 

(2)Core temperature

Core temperature can directly determine whether the heat balance is damaged. Compared with skin temperature, it is more stable and keeps constant in a certain range. Rectal and stomach temperatures are usually used to represent the body’s core temperature. In the thermal environment, the physiological upper limit of the core temperature is 38 °C, and the safety limit is 38.9 °C. It is difficult to measure the core temperature, so the skin temperature is used to estimate. There is a linear relationship between core temperature and skin temperature; the former is 2 °C to 4 °C higher than the latter [16].

(3)Reaction ability

Two indexes, choice reaction time and the number of wrong choices, were used to evaluate drivers’ response ability. The instrument LJ910XB was used to test these indexes. Test details refer to previous studies [17]. This test requires participants to judge the red, yellow, and blue lights on the screen and use their left hand, right hand, and right foot to react to these colors. Their reaction time and the number of wrong choices are automatically recorded. Each participant practices eight times first and then is formally tested 16 times. If the reaction time is less than 620 ms, their selection accuracy is considered to be high; if the reaction time is 630–980 ms, their selection accuracy is considered to be normal; if the reaction time is 990–1340 ms, their selection accuracy is considered to be poor; if the reaction time is greater than 1350 ms, then their selection accuracy is considered to be very poor. The number of wrong choices is also used to measure their selection accuracy. If the number of errors is 0, then the selection accuracy is considered to be good; 1–4 is average; 5–6 poor; greater than 7 is very poor. These two factors are used to assess the agility of the participants. 

### 3.2. Procedure

Twenty-four drivers of three bus lines were selected for the test, and the running time from the origin to the terminal of each line was about 90 min. During the test, the air temperature was 28–29 °C, 31–32 °C, 33–34 °C, the air humidity was 40–60%, the wind force was 2–3, and the compartment temperature was 28–32 °C, 32–36 °C, 37–41 °C, respectively. The drivers wore a smart wristband to monitor their heart rate and skin temperature. The reaction ability was tested immediately after the buses reached the terminal. All the drivers who took part in the test were male, aged between 40 and 55 years, and had driving experience of 5–10 years. The participants reported that they were in good health and had no disease history.

## 4. Result Analysis

### 4.1. Effects of Temperature and Driving Time on Physiological Stress Response

The average body temperature, heart rate, and *PSI* of bus drivers in different driving environments are shown in Table 3, Table 4 and Table 5.

Under the same driving time (20 min, 60 min, 80 min), the higher the compartment temperature was, the higher the body temperature was, and there were significant differences between different temperatures.

At the same compartment temperature, body temperature increased with the driving time. At 32 °C and 36 °C, the body temperature remained stable, varying from 37 °C to 37.9 °C, which indicates that heat balance can be achieved by sweating, convection, and radiation; at 40 °C, with the increase in driving time, the body temperature increased continuously, and it reached 38.2 °C by the end of driving, which indicates that the regulation mechanism can no longer play a role. The body heat accumulation leads to a state of stress.

At the same temperature, the heart rate increased with the driving time. The higher the air temperature, the more rapid the increase in heart rate. At 40 °C, the heart rate at the end of driving was 109 bpm, while at 32 °C, the heart rate was only 89 bpm. 

Under the same driving time (20 min, 40 min, 60 min, 80 min), the heart rate at different temperatures revealed significant differences. After driving for 20 min and 40 min, the average heart rate fluctuated little, and the maximum value was 98; after 60 min, the heart rate fluctuated greatly, and it exceeded 100 at 40 °C. 

Under the same driving time, the *PSI* increases with the increase of air temperature, and there are significant differences among different air temperatures. 

At the same temperature, with the increase in driving time, *PSI* also increases, and the maximum increase occurs in driving for 20–40 min, with an increasing range of 42–47%. After 40 min, the increased range clearly decreases. 

There is no stress reaction at 32 °C and 36 °C. At 40 °C, driving for 60 min, *PSI* reaches 2.751, close to the physiological stress level. After driving for 80 min, *PSI* reaches 3.316, which reaches moderate stress.

### 4.2. Effect of Temperature and Driving Time on Behavioral Response

#### 4.2.1. Statistical Analysis

The reaction times and the number of choice errors under different air temperatures are shown in Table 6.

The higher the air temperature, the slower the reaction rate. The average reaction time at 32 °C, 36 °C, and 40 °C was 724 ms, 817 ms, and 870 ms, respectively, and the reaction time at different air temperatures had significant differences.

The average number of choice errors at 32 °C, 36 °C, and 40 °C were 167, 2.750, and 2.958, respectively. There were significant differences between 40 °C and 32 °C and between 36 °C and 32 °C, indicating that judgment ability decreases with the increase of air temperature. However, there was no significant difference between 40 °C and 36 °C, indicating that the drivers are continuously affected by the temperatures under high temperatures.

#### 4.2.2. The Relationship between Physiological Stress and Behavioral Response

Physiological stress can lead to changes in behavioral responses. Pearson’s correlation analysis was used to test the relationship between them. The results are shown in Table 7.

The correlation coefficient was between 0.418–0.660, which indicated that *PSI* strongly correlated with reaction ability. In addition to 36 °C, *PSI* was positively correlated with reaction time and the number of choice errors at other temperatures, which indicated that the increase in air temperature would lead to the increased reaction time and the number of choice errors.

### 4.3. Safe Driving Time

According to the definition of physiological limit by WTO and the labor classification standard of the International Labor Bureau in 1983, the core temperature of working in high-temperature environments cannot exceed 38 °C, and the upper limit of the heart rate is 125 bpm. When one of the two indexes exceeds the physiological limit, the driver’s driving time is defined as safe driving time. This can be ascertained by counting the driving time of the all respondents when rectal temperature or heart rate reached the physiological safety limit at 32 °C, 36 °C, and 40 °C, and taking the average time as the safe driving time under different temperatures (Figure 1).

### 4.4. Survival Analysis 

The Kaplan–Meier survival function is suitable for survival analysis with a single factor influence. It allows a grouping variable to compare survival rates between groups and allows a hierarchical variable.

The test results of the difference in survival rates between groups are shown in Table 8. The results of the three test methods were consistent, indicating significant differences in the distribution of survival time under different temperatures. 

As can be seen from the survival curve in Figure 2, the higher the temperature, the lower the survival rate. When the air temperature reached 32 °C, all drivers did not reach the physiological safety limit after driving for 80 min. When the air temperature reached 36 °C, after driving for 52 min, some drivers’ physiological indexes exceeded the standard, and the survival rate decreased to 80%; when driving for 73 min, it dropped to below 60%. When the temperature reached 40 °C, the survival rate decreased to 20% after driving for 53 min; when driving for 75 min, all drivers’ physiological indexes exceeded the standard, and the survival rate was 0. At this time, the driver’s stress response ability was weak, the choice reaction time was prolonged, and the number of wrong choices increased. Driving under this condition increased the driver’s physiological risk and collision risk.

## 5. Discussion

The heat produced by the muscles during work in hot environments is not easy to dissipate, which poses the greatest challenge to normal thermoregulation [18]. The development of hyperthermia induces a supraspinal failure in sustaining neural drive during prolonged muscle contractions [19]. The study found that, at 32 °C and 36 °C, with the change in driving time, the body temperature remained stable and did not exceed the safe temperature. At 40 °C, with the increased driving time, the body temperature increased continuously. After driving for an average of 53 min, it reached the safe limit. It was found that hyperthermia from exertion and environmental conditions during agricultural work manifests itself by various symptoms and may lead to death. From 1992 through 2006, 68 workers employed in crop production and related services died from heat-related illnesses. The crop worker fatality rate averaged four heat-related deaths per one million workers per year—20 times higher than the 0.2 rate for US civilian workers overall [20].

All the drivers who participated in the test have more than five years of driving experience and have certain heat acclimation under high temperatures. At 40 °C, sweating is faster and can lead to dehydration, which leads to heat stress [21,22]. The hysteresis phenomenon appears when subjects move from a neutral to a warm environment with psychological influence occurring after ascents and descents [23].

The heart rate increased with the increase in driving time. At 32 °C and 36 °C, the change range of the heart rate is relatively gentle, indicating that a dynamic balance can still be established inside the body to keep the core temperature stable. At 40 °C, the heart rate increased sharply in the first 40 min and then increased slowly thereafter. During the beginning of the test, the body’s heat production increased significantly, and the blood flow also increased, which led to a rapid increase in heart rate. After a period, the body gradually adapted to the thermal environment, so the heart rate increased more slowly. Studies have shown that the heart rate of workers with moderate and severe intensity increased significantly in the first 15–30 min of the hot test, and then increased slightly [24].

*PSI* is consistent with the change law of core temperature and heart rate, which can represent heat balance and the change in exercise intensity. This indicates that *PSI* can evaluate the level of heat stress, which is consistent with some studies [25,26]. The labor intensity of bus drivers is small, and the heart rate changes little, so the *PSI* index value is small. At 40 °C, bus drivers had stress reactions, and the *PSI* index reached 3.3. This result is similar to previous studies. It was found that when the wet bulb temperature is about 34 °C, the average value of *PSI* is 2.6 and 5.2 under low and high environmental stress, respectively [27]. One study found that monitored heat-acclimatized workers had an average *PSI* of 2.6 at 38.4 °C [28]. With the increase in air temperature, the driver’s reaction ability clearly decreased. The decrement in reaction ability may be at least partly related to increases in core temperature and dehydra tion. Some studies have shown that the increase in core temperature can cause nervous system disorders, resulting in poor motor function and irritability [13,29]. Under hot conditions, spatial abilities on memory tests for pattern and spatial recognition are significantly reduced, while exertional heat stress also impairs simple cognitive functions [30,31]. In high-temperature conditions (45 °C), the subject experienced increased thermal stress and exertion, even though he decreased his work output and employed aggressive fluid replacement [32]. An imprecise but positive relationship exists between climate change and occupational heat strain in outdoor workers, and the most likely mechanism involves dehydration, fatigue, dizziness, confusion, reduced brain function, loss of concentration, and discomfort [33,34]. Some studies have concluded that rain and high temperatures do not increase the risk of bus accidents, which is inconsistent with the conclusions of this study. However, it does not negate the impact of hotter-than-usual weather on bus accidents [35].

The safe driving time of bus drivers at 32 °C, 36 °C, and 40 °C was 80 min, 73 min, and 53 min, respectively. These safety times are longer than existing standards [36,37]. One possible reason is that the influence of humidity was not considered. Second, the wind can reduce the driver’s body surface temperature because the windows are open. In addition, the driver can relax properly during the boarding and alighting of passengers.

Studies have investigated drivers’ traffic stress by filling out the Driver Stress Inventory (DSI) and demographic information sheets. This study assessed the driver’s physiological stress by measuring the driver’s heart rate, core temperature, and reaction ability. This method can be more objective and intuitive [38].

## 6. Measures to Reduce Stress Response

Several mechanisms of heat transfer from the body to the environment include convection, conduction, and behavior (e.g., increase ventilation, drink water, or modify envi ronmental controls) [39].

### 6.1. Hydration

The ability of prolonged moderate intensity exercise in a hot environment is adversely affected by dehydration, which may be associated with decreased sweat secretion and increased rectal temperature and heart rate. If the total heat load and sweat rate are high, it is more and more difficult to replenish the water lost in sweat. The mechanism of thirst is usually not enough to drive one to drink large amounts of water to replenish the water lost in sweat [40]. Thus, drivers should take the initiative to replenish water to slow the increase in rectal temperature and reduce the stress response. NIOSH recommends drinking a glass of water every 15–20 min to prevent dehydration. However, we should also pay attention to the fact that the drivers can not go to the toilet during driving, so the amount of drinking water should be determined according to the actual situation of the drivers.

### 6.2. Convective 

Dry bulb air temperature (*t**_a_*) and air velocity(*v**_a_*) are the variables that determine the convective heat exchange between the driver and the vehicle's interior environment. When air temperature (*t**_a_*) is higher than the mean skin temperature (*t**_sk_* of 35 °C), heat is gained by convection. When *t**_a_* is less than *t**_sk_*, increasing air movement across the skin by increasing local ventilation will increase the rate of body heat loss. In addition, as long as *t**_a_* exceeds *t**_sk_*, *v**_a_* should be reduced to a level where sweat can still evaporate freely, but the convective heat gain will be reduced. Therefore, in order to increase local ventilation, we can consider adding fans to buses. The cost of the fan is low, and the circulating air can effectively reduce the thermal environment around the driver. The installation position of the fan does not affect driving safety and can be installed on the driver’s head or side.

### 6.3. Work Schedule 

Special scheduling should be set up in hot weather by dividing the day’s driving time into several periods and arranging the working time according to the driver’s physiological index and driving experience [41]. After returning to the terminal, the driver should take a rest and recover. The following work can only be arranged after meeting the physiological indicators.

## 7. Conclusions

By measuring the heart rate, body temperature, and reaction ability parameters of bus drivers under different temperature environments, this paper analyzed the changes in *PSI*, the relationship between *PSI* and reaction ability, safe driving time, and survival rate under different temperatures and driving times. The study found the quantitative results of safe driving time and survival rate of bus drivers in different environments, which has important guiding significance for improving the driving environment of bus drivers and reasonably arranging working hours. However, there are some limitations to this study.

First, there are no female drivers in the sample, which may affect the average safe driving time. The research shows that the physiological and psychological changes of drivers of different genders in high temperatures are different [42,43]. Secondly, the air humidity during the survey period was small, and the influence of humidity was not considered. Even if the temperature is comfortable, high humidity could result in heat stress and possible heat injury to workers [39]. In some areas, high temperature is often accompanied by high humidity. It is necessary to study the driver’s changes under different temperature and humidity combinations in the future. Finally, because it is an actual vehicle environment test, it is impossible to test the driver’s reaction ability in each period.

## Figures and Tables

**Figure 1 ijerph-19-09662-f001:**
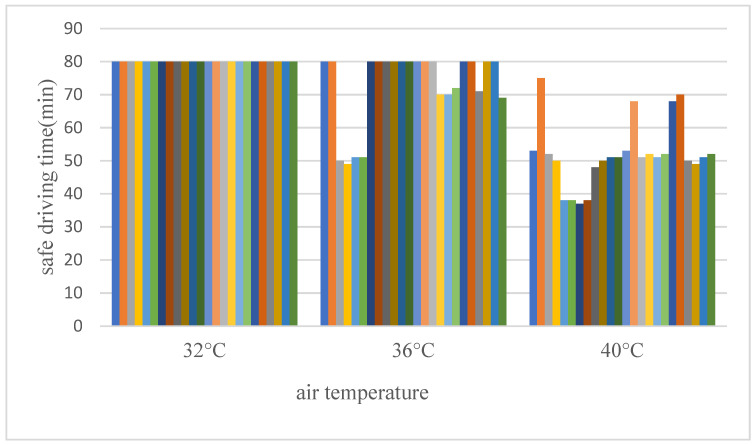
Safe driving time under different temperatures.

**Figure 2 ijerph-19-09662-f002:**
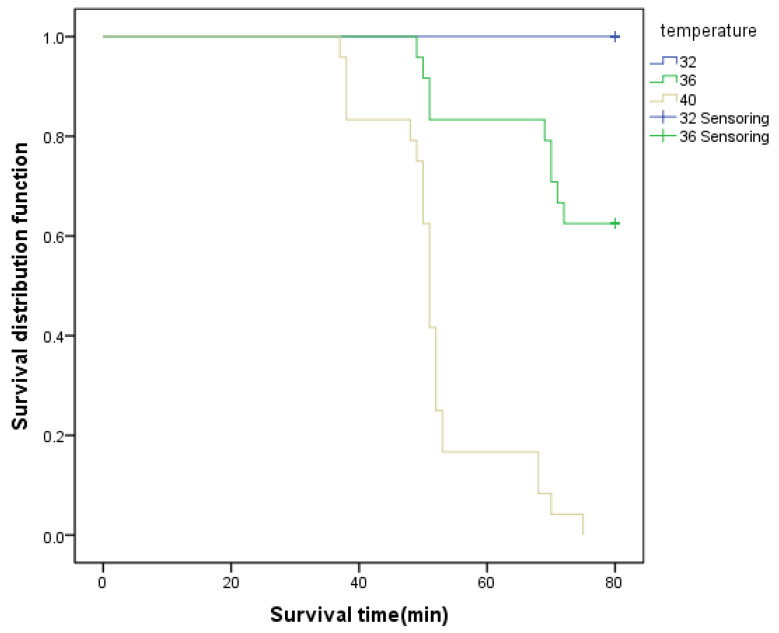
Survival curve of bus drivers in different temperature environments.

**Table 1 ijerph-19-09662-t001:** Classification standard of labor intensity.

Labor Intensity	Medium	Heavy	Very Heavy
*HR*	100–125	125–150	150–175
*RT*	37.5–38	38–38.5	38.5–39

**Table 2 ijerph-19-09662-t002:** Classification standard of *PSI*.

Labor Intensity	Medium	Heavy	Very Heavy
*PSI*	2.8	2.8–5.6	5.6

**Table 3 ijerph-19-09662-t003:** Bus drivers’ body temperature under different temperatures and driving times.

Compartment Temperature	Continuous Driving Time							
	20 min		40 min		60 min		80 min	
	Mean	SD	Mean	SD	Mean	SD	Mean	SD
32 °C	370.0	0.162	370.2	0.164	370.3	0.167	370.5	0.228
36 °C	370.5	0.139	370.7	0.178	370.8	0.197	370.9	0.147
40 °C	370.7	0.167	370.9	0.171	380.1	0.153	380.2	0.134
	F = 107.819 * *p* = 0.000		F = 1.076 *p* = 0.402		F = 116.13 * *p* = 0.000		F = 106.137 * *p* = 0.000	

* *p* < 0.05.

**Table 4 ijerph-19-09662-t004:** Bus drivers’ heart rates under different temperatures and driving times.

Compartment Temperature	Continuous Driving Time							
	20 min		40 min		60 min		80 min	
	Mean	SD	Mean	SD	Mean	SD	Mean	SD
32 °C	82	2.938	85	3.519	87	4.138	89	4.856
36 °C	86	3.659	90	4.836	93	6.271	94	6.021
40 °C	91	4.785	98	6.828	104	5.987	109	6.617
	F = 34.33 * *p* = 0.000		F = 37.818 * *p* = 0.000		F = 56.913 * *p* = 0.000		F = 71.168 * *p* = 0.000	

* *p* < 0.05.

**Table 5 ijerph-19-09662-t005:** Bus drivers’ *PSI* under different temperatures and driving times.

Compartment Temperature	Continuous Driving Time							
	20 min		40 min		60 min		80 min	
	Mean	SD	Mean	SD	Mean	SD	Mean	SD
32 °C	0.654	0.276	1.145	0.264	1.497	0.356	1.887	0.520
36 °C	0.814	0.231	1.405	0.398	1.858	0.502	2.114	0.441
40 °C	1.086	0.310	2.055	0.470	2.751	0.453	3.316	0.618
	F = 15.217 * *p* = 0.000		F = 35.167 * *p* = 0.000		F = 51.363 * *p* = 0.000		F = 50.138 * *p* = 0.000	

* *p* < 0.05.

**Table 6 ijerph-19-09662-t006:** Driver’s reaction ability at different temperatures.

Temperature	Reaction Time (ms)		Number of Choice Errors	
	Mean	SD	Mean	SD
32 °C	723.833	18.851	2.167	0.637
36 °C	816.875	18.197	2.750 ^a^	0.531
40 °C	869.583	21.518	2.958 ^b^	0.690
	F = 341.119 *, *p* = 000		F = 10.409 *, *p* = 000 (a–b)^ns^, *p* = 0.251	

ANOVA and post hoc tests with LSD correction were conducted with the level of significance at 0.05. Multiple comparisons: (a,b)^ns^ denote no significant difference between groups a and b. * *p* < 0.05.

**Table 7 ijerph-19-09662-t007:** Relationship between physiological stress and reaction ability.

Physiological Stress	Reaction Time	Number of Choice Error
*PSI* _32_	0.606 *, *p* = 0.002	0.458 *, *p* = 0.024
*PSI* _36_	0.499 *, *p* = 0.013	0.330, *p* = 0.116
*PSI* _40_	0.418 *, *p* = 0.042	0.660 *, *p* = 0.000

* *p* < 0.05.

**Table 8 ijerph-19-09662-t008:** overall comparison.

Test Method	*X* ^2^	*df*	sig.
Log Rank (Mantel-Cox)	72.110	2	0.000
Breslow (Generalized Wilcoxon)	62.128	2	0.000
Tarone-Ware	67.039	2	0.000

## Data Availability

The data used in this study are available through the corresponding author upon request.

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
