# Peer review of "Stress Response and Safe Driving Time of Bus Drivers in Hot Weather"

_ijerph, 2022, doi:10.3390/ijerph19159662_

Round 1
Reviewer 1 Report
This paper analyzed the relationship between working environment and driving safety. They authors found that working in a high temperature for a long time could increase the heart beat and body temperature, resulting in the reduction of safe driving time. 24 drivers participated in the naturalistic driving experiments. The results were easy to understand and sound reasonable.
Some comments:
1. Paper formatting issue, space between the word '36C' and 'when' in Abstract.
2. Expression error, “when one of the two indexes of the interviewees exceeds the safety physiological limit, the driving time is safe driving time” in 4.3 Safe driving time section.
3. Drivers' age are all between 40 and 55 years, whether the age has relation with rectal temperature and heart rate? Because young drivers’ body performance may be different from older drivers.
4. Using skin temperature(wrist) to estimate the core temperature should be further proved, why not forehead temperature?
5. As mentioned in Measures to reduce stress response, drinking water plays a necessary role. Do you ensure that the driver did not drink water during the experiment?
6. From the referee’s knowledge, the heterogeneity in driver’s response to working environment should also be a non-negligible variable.
Author Response
Thank you very much for your comments and suggestions regarding our manuscript. The manuscript has been carefully revised.

Reviewer 2 Report
Interesting article, but it should be revised.
Because:
(a) the references on page 11 are not all cited in the text, so the organization of the article should be revised and taken into account. In the article, there are no references to the numbers: 4, 12, 15, 17, 18, 19, 26, 27 and 28.
b) We think that topic 3.1 (measurement method) should be integrated in topic 2 and not as a title (3. Data), because the authors are not dealing with the data here. Thus, the first subtitle of 3.1. Data should be 3.1. Procedures and not 3.2.
c) Point 6 (page 8) is misplaced. It should come before the discussion of the data and the conclusion and not between these two topics. Also and in the conclusion they could refer to these topics as a way to reduce the stress response.
d) on page 8 line 255 should start with a capital letter at the beginning of the sentence. The same happens at the end of line 54 (page 2).
Author Response

(The authors gave the same response as above.)

Reviewer 3 Report
Although I find the content interesting and useful to a wide population- it was very difficult to read with extensive editing required to make it easier on the reader to follow. An example of confusing sentences is the sentence starting in line 251. And then after that, the entire next paragraph is an example of an area needing heavy editing for reader ease.
In addition, the methods are not clearly described. I feel like a short description of reaction time and number of choice errors would help understand the methods a bit more instead of just referring to another research paper.
Some of the Discussion seemed much more detailed in some areas unnecessarily than the rest of the paper and a bit out of place (line 227)
I particularly liked section 6 and feel that is some information that needs to be disseminated to stakeholders.
Author Response

(The authors gave the same response as above.)

Round 2
Reviewer 3 Report
I find this paper interesting, but it is so difficult to follow in English that I think it does more harm to the cause than good. On my first review, I kindly noted that the language had to be greatly improved and I see very little improvements.
Author Response
Thank you very much for your comments and suggestions regarding our manuscript. We have asked professionals whose mother tongue is English to revise the language of the paper. The revised paper is attached.
